# Piperine Derived from *Piper nigrum* L. Inhibits LPS-Induced Inflammatory through the MAPK and NF-κB Signalling Pathways in RAW264.7 Cells

**DOI:** 10.3390/foods11192990

**Published:** 2022-09-26

**Authors:** Zhouwei Duan, Hui Xie, Shasha Yu, Shiping Wang, Hong Yang

**Affiliations:** 1College of Food Science and Technology, Huazhong Agricultural University, Wuhan 430070, China; 2Institute of Agro-Products Processing and Design, Hainan Academy of Agricultural Science, Haikou 571100, China; 3College of Food Science and Technology, Hainan University, Haikou 570228, China

**Keywords:** *Piper nigrum* L., anti-inflammatory, piperine, MAPK, NF-κB

## Abstract

Piperine, an important natural product, has a good anti-inflammatory effect. However, few researchers have studied its mechanism in these pathways. The objective of this research was to evaluate the molecular mechanism underlying the anti-inflammatory responses of piperine in lipopolysaccharide (LPS)-stimulated RAW264.7 cells. The purification and characterization of piperine from *Piper nigrum* L. were determined by HPLC, UPLC-Q-TOF-MS and ^1^H NMR. Then, the anti-inflammatory activity was evaluated by a reagent test kit, ELISA kits, RT-PCR and Western blot experiments. The results suggested that piperine (90.65 ± 0.46% purity) at a concentration of 10–20 mg/L attenuated the production of NO and ROS, downregulated the protein and mRNA expression levels of TNF-α, IL-1β and IL-6, and upregulated the protein and mRNA transcription levels of IL-10. Meanwhile, the Western blot results indicated that piperine could inhibit the phosphorylation levels of the ERK, JNK, p38 and p65 proteins. Our findings suggest that piperine is a potential anti-inflammatory substance, whose molecular mechanism may be to regulate the key factors of the NF-κB and MAPK signalling pathways.

## 1. Introduction

*Piper nigrum Linn*, belonging to the Piperaceae family, is mainly cultivated for its fruits in Malaysia, Vietnam, China, and other countries [1]. *P. nigrum* is a traditional dual-use resource of medicine and food resources in China, while traditionally it can be used for treating chills, stomach, detoxification and rheumatism [2]. Piperine is the major active compound in both white and black pepper. Several studies have indicated that piperine has various biological functions, including immune enhancement [3], anti-cancer [4], anti-inflammatory [5], antioxidant [6], and antibacterial activities [7,8]. Due to its biological activity, the application of piperine has increased globally in the past couple of years.

Inflammation is the natural defence mechanism of the human body in response to various inflammatory factors [9]. Moderate inflammatory reactions are beneficial to body health, but excessive and persistent inflammatory responses can cause cancer and autoimmune liver disease [10]. Recently, an increasing number of natural compounds have been identified as anti-inflammatory compounds to help avoid potential harm to human health. Natural compounds can potentially reduce inflammation and have multichannel, multilevel and high safety. The application of natural compounds to attenuate inflammation has become a current research hotspot [10]. Agricultural products are regarded as a natural treasure trove of “natural products”. It is necessary to find nutritional components from natural food raw materials to safely and effectively attenuate inflammatory reactions.

The effects of piperine on immunomodulation were mainly focused on the immune protection of intestinal epithelial cells. Intestinal epithelial cells and macrophages can regulate immunity via multiple response mechanisms [11]. Piperine (99.2% purity) attenuated the inflammatory response by suppressing the secretion of interleukin-8 (IL-8) in HT-29 and SW480 cells [12]. Piperine can suppress NO and PGE2 secretion in lipopolysaccharide (LPS) activated RAW264.7 cells [13]. Despite these beneficial results, no study has yet evaluated the anti-inflammation effects of piperine, which is extracted and purified from white pepper in the Hainan Province of China. Different sources, components, purity and treatment of piperine exhibited different biological activities. Inflammation regulated by macrophages has an important effect in human disease. However, few studies have examined the effect of piperine on macrophage inflammation regulation. The signalling pathways by which piperine from *Piper nigrum* L. exerts anti-inflammatory activity remain unclear.

LPS is the main component of the cell wall of gram-negative bacteria [14]. It can create an inflammatory response by activating the cellular inflammatory signalling pathway, causing a cascade reaction, and secreting inflammatory factors [15]. Macrophages contribute to the innate immune response to defend against inflammatory diseases and pathogen infection. Therefore, LPS-induced RAW 264.7 cells are usually used as a model to study the mechanisms involving anti-inflammatory effects [16,17]. Mitogen-activated protein kinase (MAPK) and nuclear factor kappa B (NF-κB) is closely related to inflammatory responses [18]. Our previous study isolated and purified piperine from *Piper nigrum* L., and it exhibited immunomodulatory capacity. Meanwhile, the regulation of immune responses is closely related to anti-inflammatory effects. The objective of this study was to establish an LPS-stimulated RAW264.7-cell model to evaluate the anti-inflammatory action. Subsequently, the potential molecular mechanisms of piperine in the MAPK and NF-κB signalling pathway were investigated to provide a reference for the physiological activity of *Piper nigrum* L.

## 2. Materials and Methods

### 2.1. Materials and Chemicals

White pepper was purchased from Hainan Xinghuida Agricultural Technology Co., Ltd. (Hainan, China). Foetal bovine serum (FBS) and Dulbecco’s modified eagle medium (DEME) were purchased from Gibco Life Technologies (Waltham, MA, USA). Reactive oxygen species (ROS), Dimethyl sulfoxide (DMSO), Nitric oxide (NO), 3-(4,5-Dimethyl-lthiazol-2-yl)-2,5-diphenyltetrazolium bromide (MTT) Kits were purchased from Beijing Soleibo Technology Co., Ltd. (Beijing, China). Interleukin (IL)-10, IL-1β, IL-6, tumour necrosis factor-α (TNF-α) ELISA kits were purchased from Nanjing Jiancheng Bioengineering Institute (Nanjing, China). The lactate dehydrogenase (LDH) cytotoxicity assay kit was obtained from Beyotime Biotechnology CO., Ltd. (Shanghai, China). The TRIzol reagent, reverse transcription kit, 2 × SYBR Green qPCR Master Mix, β-actin, GAPDH, and antibodies against phospho-p38/p65/JNK/ERK were obtained from Wuhan Service Biotechnology Co., Ltd. (Wuhan, China).

### 2.2. Extraction and Purification of Piperine

White pepper was crushed and sieved through a 40 mesh to obtain a powder. Piperine was extracted from pepper powder with ethanol (1:25, g/mL) at 60 °C for 60 min. The extracted slurry was filtered with medium speed filter paper when extraction was performed. The supernatant was concentrated by a rotary evaporator (Yamato, Tokyo, Japan) at 50 °C under a vacuum. Then, vacuum freeze-drying was used to obtain a crude extraction of piperine. Afterwards, HPD22 resin with 2.0 mg/mL crude extract was loaded at a flow rate of 1.5 mL/min and desorbed with 90% (*v*/*v*) ethanol at a current velocity of 2.0 mg/mL. Using absolute ethanol as a solvent, the piperine purified by HPD22 resin was prepared into a slurry with a concentration of 200 mg/mL and crystallized at 4 °C for 24 h. After filtration, the precipitate was crystallized twice with absolute ethanol and then vacuum freeze-dried to obtain the purified piperine.

### 2.3. HPLC Determination of Piperine Purity

The sample (standard and extracts of piperine) was determined by HPLC, which was performed on a 1220 Infinity Ⅱ LC (Agilent, San Jose, CA, USA) with a Kromasil 100-5c C_18_ column (250 × 4.6 mm, 5 μm). The detection wavelength was 343 nm. The mobile phase was 77% methanol. The sample was dissolved in methanol, and 10 μL was drawn in volume. The standard curve was *y* = 65.425*x* + 0.1062, *R*^2^ = 0.9999, where *x* is the concentration of piperine and *y* is the peak area. The linear relationship was good in the range of the piperine solution of 0.40~2.00 mg/L.

### 2.4. UPLC-Q-TOF-MS Analysis of Piperine

The piperine was submitted to chromatographic analysis by UPLC-Q-TOF-MS and analysed on an Acquity UPLC I-Class System high-performance fluid chromatograph equipped with a Xevo G2-XS QTOF mass spectrometer, an electrospray interface (Waters, Worcestershire, MA, USA), and an Acquity UPLC BEH C_18_ column (2.1 × 100 mm, 1.7 μm). The dry piperine sample was redissolved in methanol and then filtered with a 0.22 μm nylon filter. The mobile phase used was 0.10% formic acid (A) and methanol (B). A 1 μL volume of sample was injected into the column at the temperature of 40 °C. The elution gradient of methanol was used as follows: 0–2 min, 5–10%; 2–5 min, 10–22%; 5–10 min, 22–24%; 10–13 min, 24–35%; 13–16 min, 35–45%; 16–21 min, 45–51%; 21–23 min, 51–55%; 23–25 min, 55–60%; 25–28 min, 60–70%; 28–32 min, 70–80%; 32–36 min, 80–85%; 36–40 min, 85–95%; 40–45 min, 95% and 45–48 min, 95–5%. The flow rate was 0.3 mL/min. The cone and desolvation gas flowed at 50 L/h and 1000 L/h. The ion source and desolvation temperatures were 120 °C and 450 °C, respectively. The mass scan range was from 50 to 1500 amu in positive ionization mode.

### 2.5. NMR Analysis

To determine the structure, the isolated compound was dissolved in deuterochloroform. ^1^H NMR spectra were recorded on a Bruker AV 300 MHz NMR spectrometer (Bruker, Karlsruhe, Germany). Chemical shifts are reported in δ (ppm) relative to the signal of tetramethylsilane (TMS) as the internal standard.

### 2.6. Cell Culture Conditions

RAW264.7 cells were obtained from the Chinese Academy of Sciences (Shanghai, China). The cells were maintained in DMEM that contained 10% (*v*/*v*) FBS, streptomycin (100 μg/mL) and penicillin (100 U/mL), and incubated in a humidified atmosphere (ThermoFisher, Waltham, MA, USA) with 5% CO_2_ at 37 °C.

### 2.7. Piperine Cytotoxicity Analysis

#### 2.7.1. MTT Assay

Cell cytotoxicity was evaluated by the MTT method [19]. RAW264.7 cells (1 × 10^4^ cells/mL) were plated in 96-well plates (100 μL/well). After incubation for 24 h, the new medium containing different concentrations (5, 10, 20, 40, 60, 80, 100, 120 and 140 mg/L) of piperine was replayed with the old medium, while 100 μL/well DEME medium was added to the control group. After culturing for 24 h, the medium was removed, 100 μL of 500 μg/mL MTT solution was dissolved in the medium, and the cells were incubated for 4 h in the dark. Then, the supernatant liquid was removed, and 100 μL of DMSO was subsequently added for crystal dissolution. The absorbance of the cell suspension was measured at 490 nm by a microplate reader (ThermoFisher, MA, USA).

#### 2.7.2. LDH Assay

Cell culture and grouping were the same as described in Section 2.7.1. The release rate of LDH was measured according to the LDH cytotoxicity assay kit instructions.

### 2.8. Morphological Observation of RAW 264.7 Cells

Cells were plated at 5 × 10^5^ cells/mL in 24-well plates and incubated at 37 °C for 24 h. The medium was removed, and new medium was added. Cells were divided into the control, LPS and two experimental groups. The experimental group was pretreated with 10 and 20 mg/L piperine for 2 h, while DEME medium was added to the other groups. Then, except for the control, all groups were incubated with 500 μg/L LPS solution. After incubation for 24 h, the cell morphology of the four groups was observed with an inverted light microscope (Olympus, Tokyo, Japan). The field of view of the optical microscope was adjusted at a magnification of 40×.

### 2.9. Determination of NO Content

Cell culture and grouping were the same as described in Section 2.8. After incubation for 32 h, NO production was measured according to the Griess kit instructions.

### 2.10. ROS Levels

Cell culture and grouping were the same as described in Section 2.8. After incubation for 3 h, the ROS levels were quantified with the fluorescent probe DCFH-DA according to the kit instructions. The fluorescence morphology of RAW264.7 cells was observed with a fluorescence microscope (Olympus, Tokyo, Japan).

### 2.11. Measurement of Inflammatory Cytokines

All the procedures were the same as described in Section 2.8, except cells were plated at 7 × 10^5^ cells/mL in 24-well plates with 500 μL/well. After incubation for 28 h, the culture medium was removed, and the wells were blown down with ice-cold PBS. Cell culture supernatants were centrifuged at 2000× *g* for 5 min at 4 °C. The contents of TNF-α, IL-1β, IL-6 and IL-10 were quantified by ELISA kits.

### 2.12. RT–qPCR Analysis of Inflammatory Cytokines

All the procedures were the same as described in Section 2.11, except cells were plated in 6-well plates at 2 mL/well. The cells were blown down with ice-cold PBS and centrifuged at 1500× *g* for 3 min at 4 °C. Then, total RNA samples were obtained using TRIzol reagent. Next, RNA was transcribed into cDNA under the following conditions: 95 °C for 10 min, followed by 40 cycles of 95 °C for 15 s and 60 °C for 30 s, with a dissolution temperature from 65 °C to 95 °C. The specific primers for targeted genes and GAPDH based on Rattus sequences were designed via Primer Premier software (Table 1). The volume of total reverse transcribed RNA was determined according to the concentration. The gene transcription level data were computed using the 2^−ΔΔCt^ method [20,21].

### 2.13. Western Blot Analysis

All the procedures were the same as described in Section 2.12. Then the supernatant was removed and RIPA lysis solution was added. After lysis on ice for 30 min, the supernatant (total protein solution) was collected. The protein content was quantified using a bicinchoninic acid (BCA) kit. Total proteins were separated by 10% SDS–PAGE and then transferred to polyvinylidene difluoride (PVDF) membranes. Subsequently, Western blot analysis was performed as previously described [22,23].

### 2.14. Statistical Analysis

All data are expressed as the means±standard deviation (SD). One-way variance analysis (ANOVA) was performed for statistical analysis. *p* < 0.05 was considered statistically significant. Western blot analysis was quantified by Alpha. The fluorescence intensity of ROS was measured by Image J software (NIH, Bethesda, MD, USA). SPSS 26.0 (SPSS Inc. Chicago, NJ, USA) and Origin 26.0 software (IBM, Armonk, NY, USA) were used for statistical analysis and drawing. 

## 3. Results 

### 3.1. HPLC and UPLC-Q-TOF-MS Analysis of Piperine

The chromatograms of the standard and piperine samples (described in Section 2.2) were determined by HPLC and are shown in Figure 1A,B, respectively. According to the UPLC analysis, the purity of the prepared piperine sample was 90.65 ± 0.46%. The relative molecular ion was employed for the MS evaluation of primary mass spectrometry. The sample spectrogram showed that the peak at 34.15 min was a fragment of *m*/*z* 286.1336 [M+H]^+^ (Figure 1C,D), which was close to the standard of consistency and agreed with the pattern of piperine in Yu’s report [2]. The identity of piperine was confirmed by ^1^H NMR (Figure 1E). Data for piperine: ^1^H NMR (300 MHz, CDCl_3_) δ: 7.39 (q, J = 3.0 Hz, 1H), 6.98 (s, 1H), 6.89 (d, J = 6.0 Hz, 1H), 6.77 (d, J = 12.0 Hz, 3H), 6.44 (d, J = 15.0 Hz, 1H), 5.98 (s, 2H), 3.58 (d, J = 27.0 Hz, 4H), 1.58 (s, 6H). Thus, the sample compound extraction from white pepper was identified to be piperine, and its structure is shown in Figure 1F.

### 3.2. Cell Cytotoxicity and Microscopic Morphology of RAW264.7 Cells

Cell viability reflected the extent to which cells are injured by the external environment. As shown in Figure 2A,B, when the mass concentration of piperine was less than 40 mg/L, piperine showed no significant influence on cell viability (99.40–100.00% cell viability; 100.00–102.25% LDH release rate). When the mass concentration of piperine was 60 mg/L, the cell viability declined to 90.43 ± 2.11%, which was significantly lower than that of the control group. The release rate of LDH increased to 113.44 ± 2.52%, which was significantly higher than that of the control group (*p* < 0.01). Piperine displayed no cytotoxicity in RAW264.7 murine macrophages when concentrations were below 40 mg/L, which was similar to Hou’s research [12] but different from Ying’s results [13]. Different sources, components and purities of piperine could be considered the reason for these results. As a result, piperine at concentrations of 10 and 20 mg/L was chosen for future study.

An inverted microscope was used to observe the morphology of RAW264.7 cells in various treatment groups. RAW264.7 cells in the control group were round or oval in form and contained normal intracellular organelles (Figure 2B). Cells were in close contact with each other and grew in clusters. When LPS was administered, the shape of the cells changed dramatically. RAW264.7 cells exhibited significant branching and grew unique antennae as they swelled and expanded. The distance between the cells grew noticeably wider, while the cells began to differentiate rapidly. RAW264.7 cells treated with piperine exhibited many fewer branches and antennae, while most of the cells were round or oval in shape. The cell spacing was dramatically reduced compared to that in the LPS group. It was the most effective for 20 mg/L piperine on the cells. About 500 μg/L LPS successfully induced inflammation in RAW264.7 cells and produce considerable differentiation in cell shape. Treatment with 20 mg/L piperine effectively inhibited cell differentiation and alleviated the inflammatory response.

### 3.3. Determination of NO Production and ROS Levels

NO is an important typical biomarker for evaluating whether a compound has inflammatory responses. LPS alone remarkably enhanced NO production between the control and LPS groups (*p* < 0.01) (Figure 3A). When the cells were pretreated with 10–20 mg/L piperine, the NO concentration was dramatically reduced compared to that in the LPS group. As expected, the secretion of NO decreased by 39.13% after treatment with piperine (20 mg/L) (Figure 3A). The experiments indicated that piperine dose-dependently suppressed NO generation.

LPS can cause cells to generate a large amount of ROS and free radicals, disrupting the redox balance [15]. Oxidative stress causes oxidative damage to biomolecules, resulting in the discharge of a large number of cytokines and chemokines and aggravating the inflammatory response [24]. LPS alone significantly increased the ROS level between the control and LPS groups (*p* < 0.01) (Figure 3B). When the cells were pretreated with piperine at 10–20 mg/L, the ROS production was dramatically decreased in comparison with that in the LPS group. Meanwhile, after treatment with piperine (20 mg/L), the production of piperine was reduced by 63.12%. After the preincubation of cells with piperine, the fluorescence intensity of cells was much weaker than that of the LPS group but stronger than that of the control group (Figure 3C). The results showed that piperine could reduce the ROS level in cells. 

### 3.4. Measurement of Inflammatory Cytokines

Inflammatory cytokines, such as TNF-α, IL-1β, IL-6 and IL-10 are closely linked to inflammation. As expected, the contents of TNF-α, IL-1β and IL-6 in the LPS group were much higher than those in the control group (*p* < 0.01), while IL-10 had the opposite result (*p* < 0.05) (Figure 4A–D). When the cells were pretreated with 10–20 mg/L piperine, the production of TNF-α, IL-1β and IL-6 was dramatically decreased in comparison with that in the LPS group, and IL-10 had the opposite effect. Meanwhile, after treatment with piperine (20 mg/L), the TNF-α, IL-1β and IL-6 contents were decreased by 42.92%, 65.50% and 35.42%, respectively (Figure 4A–C), but the IL-10 content was increased by 195.31% (Figure 4D). This result showed that LPS significantly improved the secretion of TNF-α, IL-1β and IL-6 compared with the control. Piperine downregulated the production of TNF-α, IL-1β and IL-6 and upregulated the secretion of IL-10. According to the results, the anti-inflammatory action of piperine was closely related to the regulation of the production of inflammatory factors.

### 3.5. RT–qPCR Analysis of Inflammatory Cytokines

The corresponding mRNA transcription levels were used as representatives to assess the effects of piperine on proinflammatory cytokines. In the present work, the mRNA expression of TNF-α, IL-1β and IL-6 in the LPS group was much higher than that in the control (*p* < 0.01) (Figure 5A–C), while IL-10 had the opposite result (*p* < 0.05) (Figure 5D). When cells were pretreated with piperine at 10–20 mg/L, the mRNA transcription levels of TNF-α, IL-1β and IL-6 were dramatically decreased compared with those in the LPS group, while IL-10 had the opposite result. After treatment with piperine (20 mg/L), the transcription levels of TNF-α, IL-1β and IL-6 mRNA were decreased by 35.89%, 66.04%, and 46.87%, respectively (Figure 5A–C). When the mass concentration of piperine was 20 mg/L, the IL-10 mRNA transcription level was remarkably improved in the piperine-treated group, and the IL-10 mRNA transcription level was increased by 135.58% (Figure 5D). These results revealed that the anti-inflammatory action of piperine was related to decreasing the mRNA transcription levels of IL-1β, TNF-α and IL-6 and improving the mRNA expression level of IL-10. Therefore, the anti-inflammatory effect of piperine is closely related to the regulation of the mRNA expression of inflammatory factors. Similar results also showed that certain bioactive components could prevent the transcription of TNF-α mRNA, inhibit the synthesis and expression of IL-1β mRNA, reduce the excessive secretion of IL-6 mRNA, and promote the expression of IL-10 mRNA in macrophages, thereby reducing the inflammatory response stimulated by LPS [25].

### 3.6. Effect of Piperine in the MAPK and NF-κB Signalling Pathways

MAPK is a group of serine/threonine-specific protein kinases that are involved in the initiation of macrophage NF-κB activation after cell stimulation. In mammalian cells, there are three representative MAPK pathways: extracellular signal-regulated kinase (ERK), c-Jun-N-terminal kinase (JNK), and p38 [25,26]. The protein expression levels of p-ERK, p-JNK and p-p38 were increased dramatically upon stimulation by LPS compared to the control (Figure 6A–D), which was similar to the finding of Gao [27]. In addition, when the concentration of piperine was 10 mg/L, the effect of p-JNK decreased significantly, while the degree of p-ERK and p-p38 was not obvious. After treatment with 20 mg/L piperine, the levels of p-ERK (Figure 6B), p-JNK (Figure 6C) and p-p38 (Figure 6D) were markedly decreased (*p* < 0.05). In the p-ERK assay, the protein level increased from 0.30 ± 0.02 to 2.13 ± 0.17 after LPS induction and decreased to 1.51 ± 0.11 with 20 mg/L piperine (Figure 6B). In the p-JNK assay, the protein level increased from 0.51 ± 0.03 to 0.84 ± 0.06 after LPS induction and decreased to 0.49 ± 0.03 with 20 mg/L piperine (Figure 6C). In the p-p38 assay, the protein level in the cells increased from 0.21 ± 0.02 to 0.47 ± 0.03 after LPS induction and decreased to 0.31 ± 0.01 in the 20 mg/L piperine group (Figure 6D). According to the above data, piperine attenuated the degree of phosphorylation of proteins corresponding to the MAPK pathway.

The NF-κB signalling pathway is a typical pathway that adjusts immunity, the inflammatory response, cell proliferation and apoptosis [28]. When macrophage cells are stimulated, the combination of active p65 and proinflammatory genes leads to an inflammatory reaction [29]. Therefore, it is necessary to further determine the transcriptional activity of the representative protein p65. Western blot analysis was used to explore the regulatory effects of piperine on p-p65 in this study (Figure 6A). The expression of the p-p65 protein in the LPS group was much higher than that in the control group (*p* < 0.01) (Figure 6E). When cells were pretreated with piperine at 10–20 mg/L, the effect of p-JNK decreased significantly. In addition, in the determination of p-p65, the protein expression level of the LPS group increased from 0.21 ± 0.02 to 0.82 ± 0.03, while it was reduced to 0.48 ± 0.03 with piperine at 20 mg/L. The results indicated that piperine inhibited the expression of NF-κB p65, conclusively suppressing the activation of the NF-κB signalling pathway.

## 4. Discussion

Inflammation is a complex physiological reaction in organisms and is closely related to numerous human diseases [9]. NO, as an intercellular biologically active molecule, is related to immune regulation and participates in tissue repair, host immune recovery, and other biological responses [30,31]. It is a signalling molecule that enhances macrophage killing activity and is linked to cytolytic action [22,32]. ROS include hydroxyl radicals, superoxide radicals and their derivatives hydrogen peroxide. The detection of intracellular ROS levels can reflect the extent of oxidative damage in organisms. Excessive ROS causes oxidative stress, which induces inflammation in the body by producing oxidative decomposition products and activating inflammatory cells to secrete proinflammatory factors [33]. Therefore, an intimate connection exists between the inflammatory reaction and oxidative response. In our research, 10–20 mg/L piperine significantly reduced the secretion of NO (Figure 3A) and ROS (Figure 3B,C). Piperine probably regulated the phagocytic activity of macrophages to inhibit the production of NO and free radicals [34]. The results suggested that the anti-inflammatory property of piperine was closed to attenuate the production of NO and ROS. Ying et al. reported that piperine (10–100 mg/L) purchased from Sigma–Aldrich can suppress the production of NO [13]. Different sources, components, purity and treatment of piperine could be considered as the factors for these results.

TNF-α plays an important role in cell activation and recruitment, as the primary “messenger” in the further induction stimulation of proinflammatory cytokine production. It triggers the secretion of inflammatory factors, including IL-6 and IL-1β [35]. IL-6 can mediate the acute inflammatory response in the early inflammation stage. When it transmits inflammatory signals as a proinflammatory cytokine, IL-6 lays the groundwork for the change from the acute transformation response into the chronic inflammatory response. It is both a marker and an inhibitor of the inflammatory response [36]. IL-1β is mainly expressed by monocytes and macrophages in inflammatory lesions [37]. It is a major orchestrator of proinflammatory cytokines involved in inflammation and autoimmune diseases. IL-10 acts as a multieffector and attenuates the deterioration of inflammation in macrophages. After RAW 264.7 cells were stimulated with LPS, the TNF-α, IL-6, and IL-1β contents were increased, which was consistent with the literature reports [38,39]. The ELISA results showed that piperine (10–20 mg/L) dramatically decreased the production of TNF-α, IL-1β and IL-6 and increased the secretion of IL-10 (Figure 4A–D). Meanwhile, when cells were pretreated with 10–20 mg/L piperine, TNF-α, IL-6 and IL-1β mRNA expression were downregulated, and IL-6 mRNA expression was upregulated, indicating that piperine had good in vitro anti-inflammatory activity (Figure 5A–D). According to the above results, the anti-inflammatory effect of piperine was related to inhibiting the production of inflammatory factors, promoting the secretion of anti-inflammatory factors, and reducing oxidative damage. Similarly, Wang’s report showed that the anti-inflammatory property of surface-layer proteins may reduce the production of IL-1β, TNF-α and ROS [40].

MAPK and NF-κB adjust many cellular processes, including inflammatory reactions, oxidative stress, cell apoptosis, and immune responses. The expression and phosphorylation levels of related proteins in NF-κB and MAPK are widely used to evaluate and screen anti-inflammatory active substances. MAPK is a major regulator of inflammatory factor gene expression and protein secretion [22,41]. After stimulation by external factors, they caused phosphorylation of ERK, p38 and JNK in the MAPK family. In the current study, ERK, JNK and p38 phosphorylation levels were enhanced in LPS-treated cells [42]. After treatment with piperine, the protein expression levels of p-ERK, p-JNK and p-p38 were dramatically decreased (Figure 6A–D). Phosphorylated proteins are transferred to the nucleus and dephosphorylation, which might activate the expression of corresponding downstream proteins, thereby regulating the release of various inflammatory factors. In conclusion, these results revealed that piperine could suppress the inflammatory reaction via inhibition of the MAPK signalling pathway.

Numerous studies have reported that LPS-mediated promotion of MAPK promotes the activation of the subsequent NF-κB and enhances cytokine expression [40]. NF-κB is a nuclear transcription factor found in a wide range of cells, where it is generally found in the form of IκB and NF-κB complexes. Studies confirmed that NF-κB is critical to the mediation of proinflammatory cytokines in the process of inflammation and immunity [43]. In this study, the phosphorylation level of p65 was remarkably enhanced in LPS-stimulated cells, indicating that LPS activated the NF-κB pathway (Figure 6A,E). After cells were induced by LPS, IκB was phosphorylated and degraded. Next, the extranuclear p65 protein was transferred from the cytoplasm into the nucleus, and NF-κB changed from a complex to a free state, allowing the NF-κB pathway to be activated. Once macrophage cells are initiated, the mute complex of IκBα/p65 begins to be ubiquitinated and degraded in the cytoplasm, and phosphorylated p65 is transferred into the nucleus and promotes the gene expression of proinflammatory factors, thus activating inflammatory responses [44]. However, the level of p-p65 was remarkably downregulated after treatment with piperine compared with that in the LPS group (Figure 6E), suggesting that piperine exhibited inhibitory activity against the NF-κB pathway.

## 5. Conclusions

This research demonstrated that piperine extracted and purified from *Piper nigrum* L. inhibited LPS-stimulated RAW264.7-cell inflammatory properties. Piperine (90.65 ± 0.46% purity) at a concentration of 10–20 mg/L decreased the production of NO and ROS. In addition, piperine attenuated the protein and mRNA transcription levels of TNF-α, IL-1β and IL-6 and enhanced the protein and mRNA expression levels of IL-10. In summary, piperine suppressed the inflammatory reaction by inhibiting MAPK and NF-κB activation, and its mechanism might be related to depressing the phosphorylation level of ERK, JNK, p38 and p65 proteins, reducing the synthesis and secretion of proinflammatory cytokines and other inflammatory mediators, and decreasing macrophage phagocytosis. These findings provide a theoretical foundation for the molecular mechanisms of the inhibitory activation of piperine in murine macrophage cells. Piperine could be used as a functional food or a new oral adjuvant for inflammatory diseases. However, the situation in vivo is complex, as the effects of compounds might be different in vivo and in vitro. For example, piperine might decompose during gastrointestinal digestion and weaken the anti-inflammatory effect. Next, the anti-inflammatory effects and pathways of piperine in vivo will be further verified.

## Figures and Tables

**Figure 1 foods-11-02990-f001:**
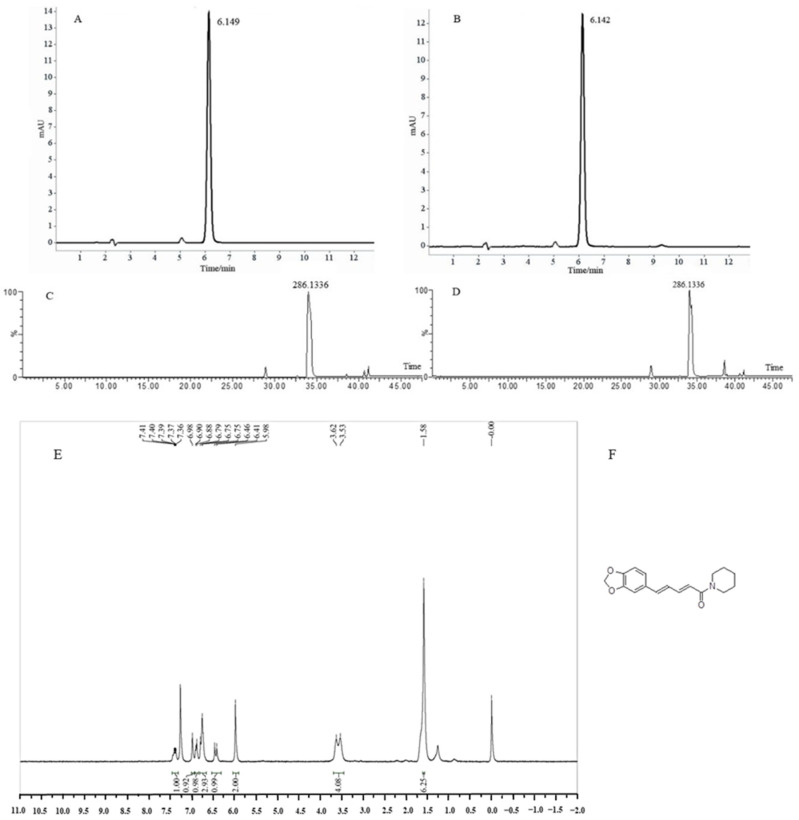
HPLC and UPLC-Q-TOF-MS analysis of piperine. (**A**) HPLC results of standards piperine; (**B**) HPLC results of sample piperine extraction from white pepper; (**C**) UPLC-Q-TOF-MS results of standards piperine; (**D**) UPLC-Q-TOF-MS results of sample piperine extraction from white pepper; (**E**) ^1^H NMR results of sample piperine extraction from white pepper; (**F**) Structural information of piperine extraction from white pepper.

**Figure 2 foods-11-02990-f002:**
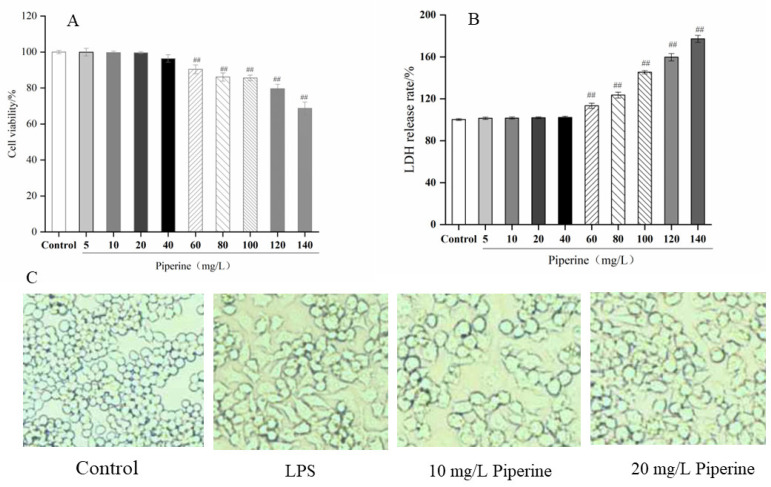
Cell viability (**A**)**,** LDH release rate (**B**) and morphology analysis (**C**) of piperine in LPS-stimulated RAW264.7 cells. Ave ± SD, ^##^
*p* < 0.01 vs. control group.

**Figure 3 foods-11-02990-f003:**
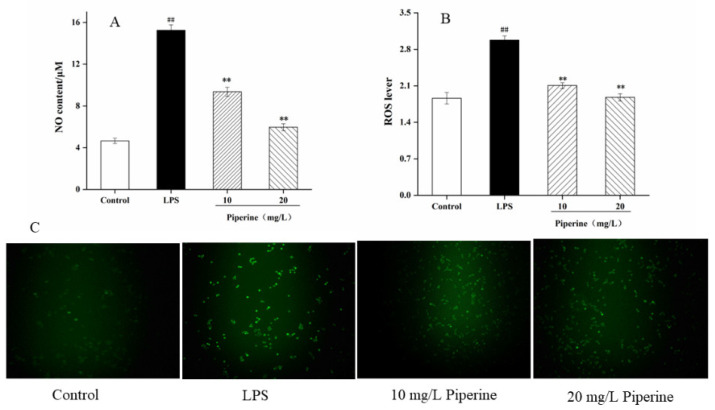
Piperine suppressed the secretion of NO and ROS. (**A**) NO secretion; (**B**) ROS lever analysis of intracellular formation; (**C**) ROS measured by Fluorescence microscopy. Ave ± SD ^##^
*p* < 0.01 vs. control group; ** *p* < 0.01 vs. LPS group.

**Figure 4 foods-11-02990-f004:**
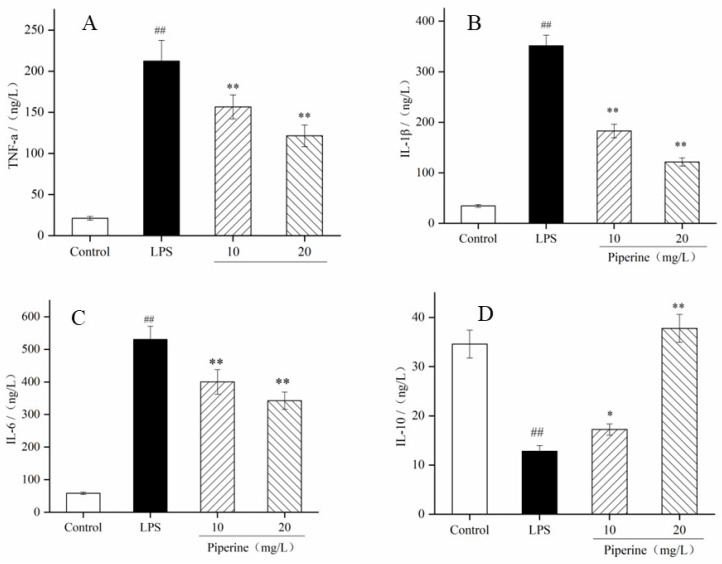
Piperine-inhibited LPS-stimulated TNF-α (**A**), IL-1β (**B**), IL-6 (**C**) and IL-10 (**D**) formation in RAW264.7 cells. Ave ± SD, ^##^
*p* < 0.01 vs. control group; * *p* < 0.05 and ** *p* < 0.01 vs. LPS group.

**Figure 5 foods-11-02990-f005:**
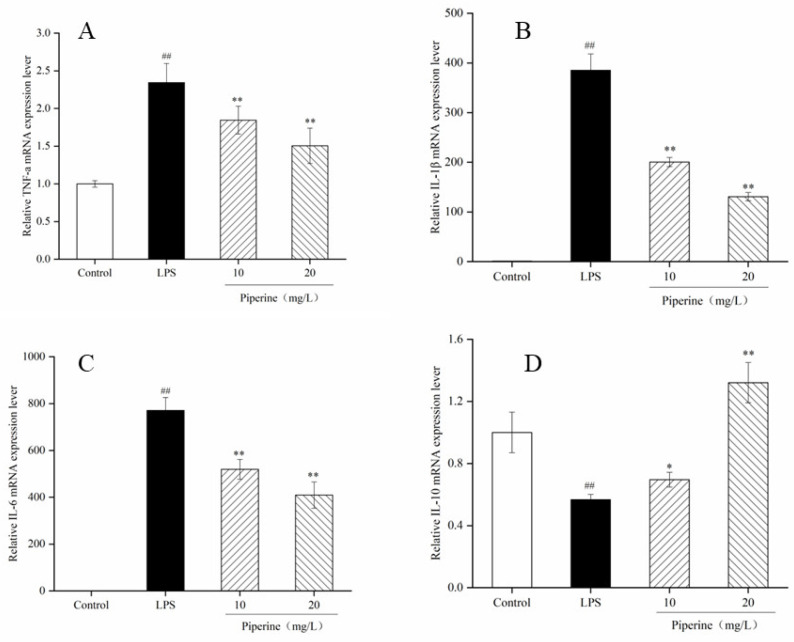
Piperine-inhibited LPS-stimulated TNF-α (**A**), IL-1β (**B**), IL-6 (**C**) and IL-10 (**D**) mRNA expression levels in RAW264.7 cells. Ave ± SD, ^##^
*p* < 0.01 vs. control group; * *p* < 0.05 and ** *p* < 0.01 vs. LPS group.

**Figure 6 foods-11-02990-f006:**
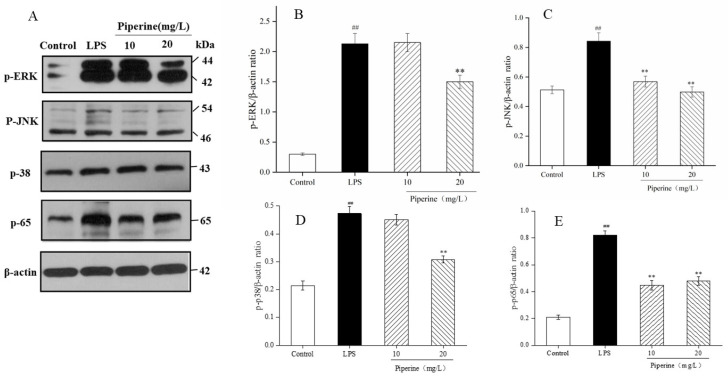
Piperine suppressed LPS-induced inflammatory activity via MAPK and NF-κB pathways. (**A**) SDS-PAGE electrophoretogram of p-ERK, p-JNK, p-p38 and p-p65. (**B**) p-ERK/ACTIN ration. (**C**) p-JNK/ACTIN ration. (**D**) p-p38/ACTIN ration. (**E**) p-p65/ACTIN ration. Ave ± SD, ^##^
*p* < 0.01 vs. control group; ** *p* < 0.01 vs. LPS group.

**Table 1 foods-11-02990-t001:** Primer information of targeted genes and GAPDH.

Gene.	Primer Sequence (5′–3′)	Orientation	Length/bp	*T_m_*/°C
GAPDH	CCTCGTCCCGTAGACAAAATG	Forward	133	60
	TGAGGTCAATGAAGGGGTCGT	Reverse		60
TNF-α	CCCTCACACTCACAAACCACC	Forward	93	60
	CTTTGAGATCCATGCCGTTG	Reverse		60
IL-1β	GCATCCAGCTTCAAATCTCGC	Forward	256	60
	TGTTCATCTCGGAGCCTGTAGTG	Reverse		60
IL-6	AGTTGTGCAATGGCAATTCTGA	Forward	229	60
	CTCTGAAGGACTCTGGCTTTGTC	Reverse		60
IL-10	AATAAGCTCCAAGACCAAGGTGT	Forward	81	60
	CATCATGTATGCTTCTATGCAGTTG	Reverse		60

## Data Availability

Data are contained within the article.

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
