# Peer review of "Piperine Derived from Piper nigrum L. Inhibits LPS-Induced Inflammatory through the MAPK and NF-κB Signalling Pathways in RAW264.7 Cells"

_foods, 2022, doi:10.3390/foods11192990_

Round 1

Reviewer 1 Report

In the current manuscript, the authors explored the anti-inflammatory potential of piperine in LPS-stimulated RAW 2647.7. The authors validated their claims through various experiments. My comments are as follows

1.  What is the novelty of the study? The anti-inflammatory activity of piperine is well established in various models including LPS-induced inflammation, how this study will add new information to the scientific literature.    2. What is DEME in the material method section?  

Author Response

Dear Reviewer:

      We are very grateful to the editor and reviewers for the effort and time spent reviewing our manuscript with the original title ‘Piperine Derived from Piper nigrum L. Inhibits LPS-induced Inflammatory through the MAPK and NF-κB Signalling Pathways in RAW264.7 Cells’. Following the comments of reviewer, we have modified our manuscript. Any revisions to the manuscript using the “*Track Changes*” function in the MS Word.

Point 1: What is the novelty of the study? The anti-inflammatory activity of piperine is well established in various models including LPS-induced inflammation, how this study will add new information to the scientific literature.

Response 1: Thank you for your suggestion.The novelty of the study as follows:(1) Piperine extracted and purified from white pepper in Hainan Province of China,which characterization was analyzed by UPLC-Q- TOF- MS and 1H NMR.(2) Piperine (90.65±0.46% purity) at a concentration of 10-20 mg/l exerted anti-inflammatory response through the MAPK and NF-κB signaling pathway. According to your suggestion, we had added some content in line 51-54.

     Ying et al. reported that piperine (10-100 mg/L) purchased from Sigma–Aldrich inhibited the LPS-mediated activation of NF-κB in RAW264.7 cells. The purity and component of piperine extracted and purified from white pepper in Hainan Province of China was different from Sigma–Aldrich. In our reaserch, piperine displayed no cytotoxicity in RAW264.7 murine macrophages when concentrations were below 40 mg/L, and at a concentration of 10-20 mg/L exerted anti-inflammatory response. Compared with piperine purchased from Sigma-Aldrich, piperine extracted and purified from white pepper showed anti-inflammatory effect through the NF-κB and MAPK signalling pathways. Different sources, components, purity and treatment of piperine exhibited different biological activities. Nutritional components from natural food raw materials may be exhibit much stronger biological activity. Our study enrich the understanding of anti-inflammatory activity of piperine and support for development of functional foods.

     Ying, X.; Yu,K.; Chen, H.; Chen, J.; Hong, S.; Cheng S.; Peng, L. Piperine inhibits LPS induced expression of inflammatory mediators in RAW 264.7 cells. Cell Immunol. 2013, 289,49-54.doi: 10.1016/j.cellimm.2013.09.001

Point 2: What is DEME in the material method section? 

Response 2: Thank you for your suggestion. According to your suggestion, to show the meaning of DEME (Dulbecco's modified eagle medium) in line 76.

Reviewer 2 Report

The manuscript titled “Piperine Derived from Piper nigrum L. Inhibits LPS-induced Inflammatory through the MAPK and NF-κB Signalling Pathways in RAW264.7 Cells” described the molecular mechanism underlying the anti-inflammatory responses of piperine in lipopolysaccharide (LPS)-stimulated RAW264.7 cells. The research design is appropriate, the results are clearly presented. However, to confirm the characterization  of piperine, the UPLC-Q-TOF-MS data was not enough, the NMR spectra must be checked. Moreover, the novelty of manuscript is rated low due to methods, research subjects, and also minor novel results. Many other research groups studied about piperine extracted from Piper nigrum L. with similar methods and objectives. For example:

Ying, X.; Yu,K.; Chen, H.; Chen, J.; Hong, S.; Cheng S.; Peng, L. Piperine inhibits LPS induced expression of inflammator medi-ators in RAW 264.7 cells. Cell Immunol2013, 289,49-54.doi: 10.1016/j.cellimm.2013.09.001

Yan Li, Kang Li, Yiqin Hu, Bo Xu, Jie Zhao. Piperine mediates LPS induced inflammatory and catabolic effects in rat intervertebral discInt J Clin Exp Pathol 2015, 8(6):6203-6213

Yi-Dan Liang, Wen-Jing Bai, Chen-Guang Li, Li-Hui Xu, Hong-Xia Wei, Hao Pan, Xian-Hui He, Dong-Yun Ouyang. Piperine Suppresses Pyroptosis and Interleukin-1β Release upon ATP Triggering and Bacterial InfectionFrontiers in Pharmacology, 2016, 7 (390), 1-12. doi: 10.3389/fphar.2016.00390.

Author Response

Dear Reviewer:

     We are very grateful to your for the effort and time spent reviewing our manuscript with the original title ‘Piperine Derived from Piper nigrum L. Inhibits LPS-induced Inflammatory through the MAPK and NF-κB Signalling Pathways in RAW264.7 Cells’. Following the comments of reviewer, we have modified our manuscript. Any revisions to the manuscript using the “*Track Changes*” function in the MS Word.

Point 1: To confirm the characterization of piperine, the UPLC-Q-TOF-MS data was not enough, the NMR spectra must be checked.

Response 1: Thank you for your suggestion. According to your suggestion, we had presented methods and results for the structure identification of piperine by NMR spectroscopy in section 2.5 (line 122-126) and 3.1 (line 202-206), respectively.

Point 2: The novelty of manuscript is rated low due to methods, research subjects, and also minor novel results. Many other research groups studied about piperine extracted from Piper nigrum L. with similar methods and objectives. For example:

    Ying, X.; Yu,K.; Chen, H.; Chen, J.; Hong, S.; Cheng S.; Peng, L. Piperine inhibits LPS induced expression of inflammator medi-ators in RAW 264.7 cells. Cell Immunol. 2013, 289,49-54.doi: 10.1016/j.cellimm.2013.09.001

    Yan Li, Kang Li, Yiqin Hu, Bo Xu, Jie Zhao. Piperine mediates LPS induced inflammatory and catabolic effects in rat intervertebral disc. Int J Clin Exp Pathol 2015, 8(6):6203-6213

   Yi-Dan Liang, Wen-Jing Bai, Chen-Guang Li, Li-Hui Xu, Hong-Xia Wei, Hao Pan, Xian-Hui He, Dong-Yun Ouyang. Piperine Suppresses Pyroptosis and Interleukin-1β Release upon ATP Triggering and Bacterial Infection. Frontiers in Pharmacology, 2016, 7 (390), 1-12. doi: 10.3389/fphar.2016.00390.

 Response 2: Thank you for your suggestion.The novelty of the study as follows:(1) Piperine extracted and purified from white pepper in Hainan Province of China,which characterization was analyzed by UPLC-Q- TOF- MS and 1H NMR.(2) Piperine (90.65±0.46% purity) at a concentration of 10-20 mg/l exerted anti-inflammatory response through the MAPK and NF-κB signaling pathway. According to your suggestion, we had added some content in line 51-54.

     Ying et al. reported that piperine (10-100 mg/l) purchased from Sigma–Aldrich inhibited the LPS-mediated activation of NF-κB in RAW264.7 cells. Yan et al. reported that piperine (10-100 mg/l) purchased from Sigma–Aldrich inhibited the LPS-mediated Nucleus Pulposus cells phosphorylation of JNK and activation of NF-κB. Yi et al. reported that piperine (20-160 μM) purchased from Guangdong Institute for Drug Control protected macrophages from pyroptosis and reduced IL-1β and HMGB1 release by suppressing ATP-induced AMPK activation in bone marrow-derived macrophages and J774A.1 cells. The purity and component of piperine extracted and purified from white pepper in Hainan Province of China was different from Sigma–Aldrich and Guangdong Institute for Drug Control.

     In our reaserch, piperine displayed no cytotoxicity in RAW264.7 murine macrophages when concentrations were below 40 mg/L, and at at a concentration of 10-20 mg/L exerted anti-inflammatory response. Compared with the piperine purchased from Sigma-Aldrich or Guangdong Institute for Drug Control, piperine extracted and purified from white pepper suppressed the inflammatory reaction by inhibiting MAPK and NF-κB activation. Different sources, components, purity and treatment of piperine could be considered as the factors for these results. Meanwhile, different cell evaluation models perhaps resulted in different results. Nutritional components from natural food raw materials may be exhibit much stronger biological activity. Our study enrich the understanding of anti-inflammatory activity of piperine and support for development of functional foods.

Reviewer 3 Report

The manuscript titled: “Piperine Derived from Piper nigrum L. Inhibits LPS-induced Inflammatory through the MAPK and NF-κB Signalling Pathways in RAW264.7 Cells” is well written. The manuscript is based on a well-constructed scientific concept, and carried out the studies are well. However, data needs to be refined in a presentable manner. The functional studies of macrophages are missing from the manuscript. The present manuscript would be benefited by addressing the points below.

Comments:

·       In figure 2, the authors have shown cytotoxicity of Piperine; authors should also show the cytotoxicity with LDH assay to quantify the cell death.  

·       Authors showed that Piperine regulates the proinflammatory phenotypes of macrophages; however, the manuscript lacks the macrophage functionality data, such as phagocytosis, and what is the effect of Piperine; below paper will be helpful to this manuscript; kindly refer to it.

Kumar VP, Prashanth KVH, Venkatesh YP. Structural analyses and immunomodulatory properties of fructo-oligosaccharides from onion (Allium cepa). Carbohydr Polym. 2015 Mar 6;117:115-122. doi: 10.1016/j.carbpol.2014.09.039.”

Author Response

Dear Reviewer:

     We are very grateful to your for the effort and time spent reviewing our manuscript with the original title ‘Piperine Derived from Piper nigrum L. Inhibits LPS-induced Inflammatory through the MAPK and NF-κB Signalling Pathways in RAW264.7 Cells’. Following the comments of reviewer, we have modified our manuscript. Any revisions to the manuscript using the “*Track Changes*” function in the MS Word.

Point 1: Data needs to be refined in a presentable manner. In figure 2, the authors have shown cytotoxicity of Piperine; authors should also show the cytotoxicity with LDH assay to quantify the cell death. 

 Response 1: Thank you for your suggestion. we had presented methods and results for the cytotoxicity with LDH assay to quantify the cell death in section 2.7.2 (line 144-146) and 3.2 (line 238, 240-241), respectively.

Point 2: The functional studies of macrophages are missing from the manuscript.

     Authors showed that Piperine regulates the proinflammatory phenotypes of macrophages; however, the manuscript lacks the macrophage functionality data, such as phagocytosis, and what is the effect of Piperine; below paper will be helpful to this manuscript; kindly refer to it.

    “Kumar VP, Prashanth KVH, Venkatesh YP. Structural analyses and immunomodulatory properties of fructo-oligosaccharides from onion (Allium cepa). Carbohydr Polym. 2015 Mar 6;117:115-122. doi: 10.1016/j.carbpol.2014.09.039.”

Response 2: Thanks for your suggestion. As the reviewer’s comments, the piperine perhaps regulate the proinflammatory phenotypes of macrophages, and the Kumar’s results showed that onion fructo-oligosaccharides could possess immunostimulatory activities towards murine lymphocytes and macrophages. It was an important roal of the macrophage functionality for proinflammatory. The anti-inflammatory effect of piperine may be regulate the phagocytic activity of macrophage or the secretion of inflammatory factor. In fact, we prefer to evaluate the potential anti-inflammatory substance of piperine whose molecular mechanism relation to the key factors of the NF-κB and MAPK signalling pathways in the manuscript. The phagocytic activity of macrophage is well investigate in Kumar´s report. Therefore, we referenced this article in the discussion section (line 414-415). Next, refer to this paper,we will further analyze anti-inflammatory effects of piperine via regulate the phagocytic activity of macrophage.

Round 2

Reviewer 2 Report

The answers and explanations of authors are clear and the revision manuscript has been improved greatly. Thank you for your effort.